# Post-Traumatic Stress Disorder in Brazilian Pregnant Women at the Beginning of the Coronavirus Disease Pandemic

**DOI:** 10.3390/ijerph21040461

**Published:** 2024-04-10

**Authors:** Jéssica Gorrão Lopes Albertini, Gláucia Rosana Guerra Benute, Maria de Lourdes Brizot, Stela Verzinhasse Peres, Rossana Pulcineli Vieira Francisco, Marco Aurélio Knippel Galletta

**Affiliations:** 1Disciplina de Obstetrícia, Departamento de Obstetrícia e Ginecologia, Faculdade de Medicina, Universidade de São Paulo, São Paulo 05508-900, Brazil; jessica.albertini@usp.br (J.G.L.A.); rossana.francisco@hc.fm.usp.br (R.P.V.F.); 2Coordination of the Psychology Course, Sao Camilo University Center, São Paulo 04263-200, Brazil

**Keywords:** post-traumatic stress disorder, COVID-19, pregnant women, state-trait anxiety, routine changes, health behavior

## Abstract

Objective: Analyze the presence of PTSD symptoms and their risk among the pregnant women during the COVID-19 pandemic. Methods: This is a cross-sectional study of pregnant women that received receiving prenatal care at two university hospitals in São Paulo, Brazil, during the COVID-19 pandemic, from April to June 2020. The sociodemographic and health data of the participants and their exposure to individuals suspected or confirmed to have COVID-19 were evaluated. The Impact of Event Scale-Revised and the State-Trait Anxiety Inventory were used to assess PTSD and anxiety symptoms, respectively. Results: A total of 149 pregnant women were included in this study. The risk of PTSD among the participants was 55.1%. The independent risk factors for PTSD were state anxiety (ORadj = 2.6), trait anxiety (ORadj = 10.7), changes in routine due to the pandemic (ORadj = 4.7) and contact with a confirmed case of COVID-19 person (ORadj = 7.1). Conclusions: The COVID-19 pandemic had a significant negative impact on the mental health of pregnant women, more than half of the participants of the present study showed a risk of PTSD, exacerbated by anxiety symptoms and exposure to individuals with a confirmed case of COVID-19.

## 1. Introduction

The presence of mental disorders during pregnancy is associated with adverse maternal and fetal outcomes, including poor adherence to prenatal care, decreased engagement in self-care behaviors, premature birth, lower birth weight, small-for-gestational-age, fetal distress, lower Apgar scores, stillbirth, and difficulties in establishing a mother–baby bond [1,2]. One of the mental health disorders that may occur during pregnancy is post-traumatic stress disorder (PTSD). PTSD is characterized by intrusive symptoms, avoidance behaviors, cognitive and mood alterations, hyperarousal, and reactivity associated with one or more traumatic events. PTSD is considered one of the most severe stress-related disorders [2], and its presence during pregnancy is associated with the occurrence of traumatic events either prior to or during pregnancy. Previous studies have indicated that the prevalence of PTSD during pregnancy among women in different countries and socioeconomic situations ranges widely from 0.6% to 16% [3]. The literature on the effects of PTSD during pregnancy on the maternal–fetal dyad indicates an association between PTSD and premature birth and low birth weight [4]. Pregnant women at increased risk of having PTSD include those with a history of violence and/or physical and sexual abuse, even in childhood; war veterans; inmates; those who abuse psychoactive substances; those with a history of pregnancy complications; and those with an intense fear of childbirth [4,5].

The coronavirus disease (COVID-19) pandemic, which began in December 2019 in Wuhan, Hubei Province, China, has introduced considerable challenges into the provision of healthcare for pregnant women. Since the start of the pandemic, there have been reports of increased rates of various mental disorders, including PTSD, among pregnant women [6], including those in Brazil. Brazil is a socially, economically, and culturally diverse country, and has a universal healthcare system with intensive care unit (ICU) beds distributed between the public and private sectors. Since the first case of COVID-19 reported on 25 February 2020, Brazil has rapidly advanced to the highest number of cases in Latin America and the highest transmission rate in the world. Plagued by worsening economic, social, political, and public health problems [7], at the end of 2020 the country accumulated 7,675,973 new cases and 194,949 deaths. The non-pharmacological measures adopted to contain the pandemic in Brazil were heterogeneous, with conflicting official guidelines on the use of masks. In the State of São Paulo, there was an official lockdown between March and August 2020, with a ban on public events and interruption of the main commercial and educational activities, remaining only the essential activities. However, adherence was not uniform throughout the region and became more flexible over time, especially for the most economically vulnerable population, who needed to work, as official emergency aid (approximately USD120 per family) was not enough nor extended to the entire population. There was also a crisis in the health system, with a shortage of hospital beds, especially for intensive care, and little availability of mechanical ventilators and oxygen. Furthermore, there was an underlying political crisis, following a heated electoral process a year earlier, which divided the popular opinion into two poles, with widespread political dissatisfaction. Data published in July 2020 revealed that the COVID-19-related mortality rate among pregnant and postpartum women in Brazil was 12.7%, which is more than three times higher than the rates reported prior to the pandemic and higher than those reported in other countries [8].

Studies on recent epidemics, such as the severe acute respiratory syndrome, Middle East respiratory syndrome, H1N1 influenza A virus, and Ebola epidemics, demonstrated that the adoption of social isolation measures to control the spread of diseases has significant psychological effects on the population, including depressive symptoms, anxiety, stress, PTSD, and emotional distress [9,10,11]. Studies published in the early stages of the COVID-19 pandemic demonstrated that social isolation measures have negative socioeconomic effects and considerably affects the physical and mental health of the general population [12], findings that were confirmed throughout 2021 and 2022.

Holmes et al. [12] indicated that it is unsurprising that social isolation and the socioeconomic effects of policies implemented to combat the spread of COVID-19 led to increased symptoms of anxiety, depression, and stress, as well as involvement in harmful behaviors such as self-harm and suicide. Fear of infection, loneliness, and a sense of confinement, possibly exacerbated by such measures, have a negative impact on mental health. Studies on the psychological effects of the COVID-19 pandemic and the measures implemented to combat them in the general population revealed several risk factors related to greater susceptibility to emotional distress during the pandemic [13]. The risk factors, which varied across studies, included sex (being female), age (being young or older than 60 years), education level (more years of schooling), history of stress or health problems (history of chronic illness), and working outside the home [9,14,15]. Ahorsu et al. [15] identified that pregnant women who had a considerable fear of COVID-19 presented with more depressive symptoms and suicidal ideation, as well as worsened mental quality of life. In a study on psychiatric symptomatology in pregnant women before and during the pandemic, which was coxnducted by Berthelot et al. [6] in Canada, participants interviewed during the pandemic showed greater emotional distress, depression, anxiety, and PTSD symptoms during prenatal care than those assessed earlier [6].

The gestational period is particularly important in the female reproductive cycle, and occurrence of mental health problems during this period can have immediate and future negative effects on the health of both the mother and her offspring. Thus, identifying the risk factors for mental health disorders in pregnant women, particularly during high-stress situations such as the COVID-19 pandemic, is crucial. Therefore, based on the above information gathered so far, we hypothesize that the COVID-19 pandemic, due to its unprecedented and acute nature, has imposed significant challenges on risk groups, such as pregnant women, in particular due to the peculiarities of the emotional experiences lived during pregnancy, with potential risk mental health in general, including impacting the perception of events as potentially traumatic, increasing the risk of developing symptoms compatible with PTSD.

## 2. Methods

This was a prospective cross-sectional study conducted at two university hospitals between 27 April 2020, and 6 June 2020. The study was approved by the Ethics and Research Committee of the institution (CAPPESQ) and registered on the National Research Registry (“Plataforma Brasil”) under the number CAEE 30298820.2.0000.0068.

### 2.1. Participants

The study participants were pregnant women, with or without clinical complications, aged 18 years or older who were receiving prenatal care at two renowned university hospitals in São Paulo, Brazil. This study was conducted during the COVID-19 pandemic, between 27 April 2020 and 6 June 2020.

The sample size was calculated using the chi-square distribution with an effect size of 0.3, α = 5%, test power (1 − β) = 80%, and degrees of freedom ≤ 5, and the results indicated that a minimum of 143 participants were required. The non-probabilistic convenience sampling method was used for analysis.

The snowball technique, which involves a chain-referral method for participant recruitment, was used for data collection. The researcher contacted key informants who helped select participants who fit the profile for the study. The participants were asked to suggest other pregnant women who could participate in the study. The survey was conducted online, and all the invited pregnant women received a link to a standardized electronic questionnaire.

### 2.2. Variables and Instruments

The sociodemographic variables collected and analyzed in this study included maternal age, gestational age (in weeks), educational level (years of education), marital status, family income, number of children residing with the participant, number of elderly individuals residing with the participant, employment status, and religious beliefs. Participants were also questioned about their previous and current health status, mental health data, presence of flu-like symptoms consistent with COVID-19 symptoms, diagnosis of COVID-19 and the time of diagnosis, contact with an individual confirmed to have COVID-19, and the perception of changes in routine after the onset of the pandemic.

The Impact of Event Scale–Revised (IES-R) was used to assess the presence of PTSD symptoms. This Likert-type scale was designed for self-administration, and respondents are required to answer the questions based on how they felt in the seven days preceding the assessment. The IES-R consists of 22 items distributed across three subscales, avoidance, intrusion, and hyperarousal, which encompass the PTSD evaluation criteria published in the Diagnostic and Statistical Manual of Mental Disorders, Fourth Edition. The score for each question ranges from 0 to 4 points, and the subscale score is determined by calculating the average of the points for items in the avoidance, intrusion, and hyperarousal subscales, excluding unanswered questions. The total score is the sum of the subscale scores. For this study, a score of ≥33 points was considered indicative of a risk of PTSD as established in a national [15] and international [16] study.

Anxiety was assessed using the State-Trait Anxiety Inventory (STAI), which was developed by Spielberger in the 1970s [17], which has been adapted in several countries and languages [18]. The objective of this scale is to measure the structural and dynamic aspects of the respondent’s anxiety. In Brazil, the STAI was translated to Portuguese and adapted by Biaggio and Natalício [19]. The test consists of two self-assessment scales for two anxiety concepts: trait and state. The trait anxiety scale assesses the individual’s personality in potentially threatening situations throughout life, whereas the state anxiety scale assesses a transient (momentary) state in which unpleasant feelings of tension and intensity depend on the situation experienced. Each scale has 20 items, and each item has four possible responses. The results are evaluated by calculating the sum of the points (maximum of 80 points) according to the criteria established by the author. The severity classification ranges from mild to moderate (20–49 points) to severe (≥50 points) [20].

### 2.3. Statistical Analysis

Descriptive analysis of the data was conducted using absolute frequency (n), relative frequency (%), measures of central tendency (mean and median), and dispersion (standard deviation and minimum and maximum values).

The chi-square test, or Fisher’s exact test when applicable, was used to assess the association between independent variables and the risk of PTSD. Univariate binary and multiple logistic regression models were used to identify odds ratios (OR) and 95% confidence intervals (CI). The stepwise backward technique was applied in the multiple regression analysis, which was conducted using the significant variables identified in the univariate logistic regression analysis (*p* < 0.050) and those with a *p* value < 0.20. Variables with the smallest observed *p* value (*p* < 0.001) up to *p* < 0.20 were entered into the multiple regression model. The model with the most precise 95% CI, variables with OR changes < 10%, and a Hosmer–Lemeshow value > 0.50 (closer to 1 being better) was chosen.

For the inclusion of variables in the multiple modeling, variables with *p*-values < 0.20 were considered, along with those related to the outcome observed previously in the literature (theoretical framework). When conducting the modeling, in the event that a variable loses its significance or changes the direction of beta, or even enhances its effects, there would be an association among them to identify potential confounding factors or interactions. Even though it was a backward modeling technique, researchers generated more than one model, and ultimately, these were evaluated by the group to ensure that, at least, the assumptions of biological plausibility and strength of association were not violated.

Statistical significance was set at *p* < 0.05. Data were analyzed using SPSS version 22.0 for Windows (IBM, Armonk, New York, NY, USA).

## 3. Results

A total of 221 pregnant women were invited to participate in this study and contacted online. Of these, 149 completed the questionnaire and were included in the study. The mean and median ages of the participants were both 32 years (SD = 6.3; range, 18–47 years). Assessment of PTSD symptoms showed that 55.1% of the participants had a risk of PTSD. The sociodemographic variables characterizing the sample are presented in Table 1.

Table 2 presents the univariate analysis of the association between gestational and health variables with the presence of symptoms compatible with PTSD during the pandemic. Variables such as gestational age and others relating to the participants’ general health status did not show a statistically significant difference between the groups with and without symptoms compatible with PTSD. However, the group of pregnant women who reported illness associated with the current pregnancy showed a statistically significant difference between the groups, where 61.8% belonged to the group with symptoms compatible with PTSD.

The data in Table 3 show that the pregnant women who received psychiatric treatment before February 2020 exhibited a higher frequency of PTSD (75.0%) than those who did not receive psychiatric treatment (51.9%; *p* = 0.054). Similarly, pregnant women with a pregnancy-related disease had a significantly higher probability (*p* = 0.018) of experiencing PTSD (61.8%) than those without a pregnancy-related disease (40.4%).

Data on state and trait anxiety, along with other variables associated with COVID-19 exposure and risk, are outlined in Table 4. We also examined the association between anxiety levels and risk of developing PTSD. The results showed significant associations between PTSD and contact with a suspected (*p* = 0.053) or confirmed (*p* = 0.001) COVID-19 patient. Pregnant women who reported changes in their routine after the start of the pandemic were more likely to develop PTSD than those who did not experience changes in their routine (59.8% vs. 17.6%; *p* = 0.001). Regarding the anxiety parameters, PTSD was associated with state and trait anxiety in the study population (*p* < 0.001).

A multiple binary logistic regression model created after univariate logistic regression analysis revealed that state and trait anxiety, changes in routine due to the pandemic and contact with a patient confirmed with COVID-19 were factors associated with the presence of symptoms compatible with PTSD (Hosmer–Lemeshow = 0.868). One of the aspects that deserves to be highlighted is the result of the trait anxiety variable, where there was a 12-fold increase in presenting symptoms compatible with PTSD (adjusted OR = 12.20; 95% CI, 4.60–32.34). Pregnant women who experienced changes in routines during the pandemic had an almost five times higher risk of developing PTSD than those who had no changes in routines due to the pandemic (adjusted OR = 4.94; 95% CI, 1.08–22.60). Pregnant women who had contact with a confirmed COVID-19 patient had an almost seven times higher risk of developing PTSD than those who had no contact with a COVID-19 patient (adjusted OR = 6.93; 95% CI, 1.07–44.69). The results are summarized in Table 5.

## 4. Discussion

The findings of this study demonstrated that 55.1% of the participants had symptoms compatible with PTSD during the initial phase of the COVID-19 pandemic. In addition, the results indicated that presence of state and trait anxiety, changes in routine due to the pandemic, and contact with confirmed COVID-19 patients were associated with the outcome symptoms compatible with PTSD.

The percentage of symptoms compatible with PTSD among the pregnant women included in this study (55.1%) is much higher than the values reported in previous studies conducted in in Brazil [21,22] and other countries [23,24,25]. A systematic review and meta-analysis by Yildiz et al. [5] indicated that the estimated global prevalence of PTSD among pregnant women before the pandemic was 4.6% [5]. The results of studies conducted during the pandemic appeared to follow a similar trend. A study conducted with Egyptian [26] pregnant women from January to December 2021 identified that 5.5% of them exhibited symptoms of PTSD. In two American studies that evaluated pregnant women during the COVID-19 pandemic, 10.3% [24] and 19.0% [27] of women presented symptoms consistent with PTSD.

Such indices are lower than those observed in the present study. Canadian researchers investigated the association between media use and the presence of symptoms of mental disorders, including PTSD, in the early stages of the COVID-19 pandemic and found that 8.58% of women exhibited symptoms of PTSD [28]. In an Italian study of 737 pregnant women [29] carried out during the first confinement in Italy, the presence of clinically significant symptoms of PTSD (assessed using the Short PTSD Scale of the National Stressful Events Survey; cut-off point: 24 points) was 10.20% [28]. The lowest rate identified to date (0.9%) was observed in a Chinese study on the prevalence of psychiatric symptoms, including PTSD, in pregnant and nonpregnant women during the COVID-19 pandemic [30].

A systematic review and meta-analysis [31] published in 2023 analyzed studies on the prevalence of mental disorders among pregnant women and those who recently gave birth during the COVID-19 pandemic, identifying a rate of 27.93% for PTSD. In a multinational study involving 5712 pregnant women aimed at evaluating factors associated with mental health, 42.8% of PTSD symptoms were identified [32].

Among the studies that have yielded results most closely aligned with our findings is that of Motrico et al. [33], which identified that, among the 3319 participants, a little over 40% exhibited symptoms of PTSD at the time of assessment. In a Turkish study conducted on pregnant women during the COVID-19 pandemic using the IES-R, present with prevalence data stratified according to symptom severity, 55.4% of the women exhibited PTSD symptoms (cutoff ≥ 33 points), which, according to the authors, indicated a moderate to severe symptom impact of the pandemic, a finding comparable with the results of the present study [34]. In another study, of Jordanian pregnant women conducted using the IES-R, which was also used in the present study, 58.6% of the pregnant women reported PTSD symptoms, a rate higher than that observed by us [35]. However, the authors used a lower cutoff score (22 points) than that used in the present study (33 points); thus, the results of both studies may not be comparable. A key point to consider when interpreting our results is the methodological differences between the present study and previous studies, specifically the instruments and cut-off scores used to identify PTSD symptoms. These disparities underscore the importance of more comprehensive studies that include, in addition to standardized instruments, clinical evaluations conducted by specialized professionals. Such an approach could provide a more accurate diagnosis of possible PTSD than relying solely on self-reporting and could guide early identification and initiation of effective intervention strategies.

When analyzing studies with higher indexes of symptoms compatible with PTSD among pregnant women, specifically those conducted in Brazil, Turkey, and Jordan, it is important to highlight the socioeconomic and political factors that constitute the context in which the COVID-19 pandemic occurred in these countries, as well as in many others. These factors include significant challenges in delivery of medical care for infected individuals, political crisis management widespread dissemination of pandemic-related news, including fake news, inappropriate management of the public health emergency situation, and economic dependence on specific sectors of the economy, such as the tertiary sector, leading to increased unemployment [8,36,37]. This complex backdrop indicates a situation that can induce anxiety, which can certainly contribute to the higher rates of PTSD symptoms in some countries and lower rates in others.

Mental health problems in pregnant women experiencing high-stress situations, such as natural disasters and various conflicts, tend to negatively affect the pregnancy and postpartum periods, as well as fetal and postpartum child development [24]. A Chinese study conducted one month after the outbreak of COVID-19 in the most affected areas indicated higher rates of PTSD symptoms among female participants than among the males (*p* < 0.01) [24]. This higher prevalence of PTSD among women could be related to hormonal fluctuations, leading to altered sensitivity to emotional stimuli and altered brain processing of fear, with increased reactivity in the neural networks involved in fear and arousal responses. Citing previous studies [9], Gómez-Salgado et al. identified that the female sex is an individual risk factor for higher levels of emotional distress during the pandemic [8]. Considering that the gestational period is characterized by numerous physical, hormonal, and social changes, experiencing PTSD symptoms during the pandemic can complicate a woman’s adaptation to pregnancy. This highlights the need for greater attention on the negative effects of the pandemic on the mental health of pregnant women.

Studies on the presence of PTSD during the perinatal period indicate that women can be exposed to numerous potentially traumatic events related to pregnancy, such as adverse obstetric outcomes, premature birth, miscarriage, clinical complications, emergency obstetric procedures, diagnosis of fetal anomalies, and low-birth-weight infants [23]. Additionally, PTSD is associated with fear of childbirth, one’s own death, and/or the death of the fetus [22,24]. Furthermore, events such as the presence of a potentially fatal illness or debilitating medical condition can be highly traumatizing.

Regarding pandemic-related gestational experiences, some differences were observed between groups with and without PTSD symptoms in the present study. More than half of the pregnant women with PTSD symptoms reported statistically significant changes in their routines after the onset of the pandemic. The present study is one of the few to investigate the association between changes in routines in the early stages of the pandemic and mental health symptoms, particularly the presence of PTSD symptoms. Masters et al. [27] identified an association between PTSD and perceived changes reported by participants; however, the authors also considered changes in access to mental healthcare [27]. This finding raises questions about how the COVID-19 pandemic changed not only how pregnant women experienced pregnancy, but also how the pandemic affected the importance of factors such as predictability of routines, personal and social organization of established routines, knowledge of prenatal care, and childbirth moments, leading to increased psychological distress.

In the present study, the percentage of pregnant women with PTSD symptoms who had contact with a suspected or confirmed COVID-19 patient was significantly higher than that of the women without PTSD symptoms. However, there was no difference in PTSD symptoms between the participants diagnosed with COVID-19 and those without COVID-19. A Spanish study in which the participants’ histories of contact with infected people and objects were assessed did not identify any association between increased psychiatric morbidity and PTSD symptoms [9]. However, other studies have demonstrated that a history of contact with a COVID-19 patient can predict increased acute stress and PTSD symptoms [24,27].

In addition to the social isolation measures imposed to curb the spread of COVID-19, the invisible threat of the risk of contamination primarily transmitted through person-to-person contact, often by asymptomatic individuals, can lead to intense fear and the perception of having been exposed to a real threat to one’s life without complete control over the process. This situation can lead to persistent memories of the risk experience or attempts to avoid similar situations in the future.

In an American study conducted with an adult population, excessive fear related to COVID-19 was clearly associated with depressive and anxiety symptoms. Excessive fear as an emotional response to an objective or subjective threat is present in the symptomatic descriptions of anxiety disorders, especially PTSD, indicating its importance in the face of a health emergency on the scale of the COVID-19 pandemic and suggesting its potential to aggravate mental distress during pregnancy [27].

The independent risk factors identified in the multiple regression analysis in the present study allow for better understanding of the process of mental health illness, not only during the pandemic but also outside it. Given that the concepts of state and trait anxiety are associated with anxiety levels that mark individual differences in an individual’s reaction to situations perceived as threatening, and can vary in intensity depending on the moment or situation, the identified risk factors may indicate that the emotional effects experienced by pregnant women during the pandemic encompass a broader and more complex process of psychological distress. Considering the risk factors identified by the model, the pregnant woman’s life history, general health history, and previous diagnosis of other mental disorders, particularly anxiety disorders, should be considered for better understanding of the emotional process leading up to the development of the symptoms and to facilitate the establishment of relevant care strategies.

Studies carried out during the COVID-19 pandemic among the general population in countries such as China [13,38], Italy [39], and Spain [40] identified that, among the sociodemographic variables, being younger, a student and female was associated with a greater risk of psychological impact related to the pandemic, which was not confirmed in the present study. The sociodemographic characteristics of the sample population of the present study may not be representative of the population of pregnant women in Brazil. Of the 150 women included in this study, 68.7% had partners, 56.7% had 9–11 years of education, and 43.3% had paid employment. This seemingly represents a population with fewer married women and higher education levels than observed in more representative Brazilian studies, such as one with nearly 10,000 pregnant women, in which 45% completed high school, 78.5% had a steady partner, and 44.2% were employed [41]. In another Brazilian study of nearly 24,000 pregnant women, 25.6% of the participants had 8–11 years of education, 81.4% had a partner, and 40.3% had paid employment [42]. However, the sociodemographic data of the participants in the present study are consistent with those reported in previous studies conducted at the same healthcare institution [32,43,44]. This indicates that the population of this study is a representative sample of our environment before the pandemic. These results could likely be extended to similar contexts.

The main strength of this study is that it presents specific prevalence data for PTSD in pregnant women during the early stages of the COVID-19 pandemic, highlighting the potential for extreme public health emergencies to negatively impact the gestational experience, destabilize pregnant women, and add mental health challenges to pregnancy, particularly when associated with abrupt changes in routine and the risk of exposure to contamination. The association between state and trait anxiety symptoms and PTSD could be a possible pathway for early identification of PTSD during pregnancy. Health professionals should pay attention to anxiety symptoms during prenatal care and consider possible traumatic experiences before or during pregnancy to identify the need for specific evaluations, expedited referrals, and initiation of treatment. In addition, the association between adverse economic environments high levels of anxiety, and the disruptive capacity of sudden changes in the routine of pregnant women, such as migration and changing jobs, even after the pandemic, should be considered.

As one of the few studies on the impact of the pandemic on PTSD symptoms in pregnant Brazilian and Latin American women, this study indicates how dangerous the lack of secure medical care for anxiety caused by lack of reliable information can be for pregnant women.

This study has some limitations. First, the cross-sectional design of the study does not allow for the establishment of causality. For instance, although the literature [45] on anxious symptoms and PTSD suggests that the presence of anxiety may increase the risk of PTSD, our findings do not allow us to make such an assertion because the association between reporting routine changes and increased PTSD symptoms may be bidirectional. That is, the presence of PTSD symptoms may influence the pregnant woman to alter her routine with the intention of avoiding situations similar to those of a traumatic event. This underscores the need for further studies to elucidate this association. Second, although PTSD is generally related to a particular traumatic event, we did not specifically investigate what each participant perceived as a traumatic event. In addition, we did not consider the fact that other traumatic events, such as previous traumatic childbirth or past or current experiences with violence, could have influenced the PTSD results. Regarding the high rate of symptoms compatible with PTSD and the possibility of comparison with other similar studies, one limitation is related to methodological and technical aspects, as there is heterogeneity regarding the instruments and cutoff scores adopted that make more precise analyses difficult.

A follow-up evaluation of PTSD symptoms in this cohort would increase our understanding of the persistence of these symptoms, creating an even more dysfunctional picture over time. New research would be necessary to verify whether such symptoms persist after the pandemic. However, understanding the factors that led to such symptoms can help improve the perception of similar cases in the future.

## 5. Conclusions

In conclusion, this study revealed that in the early stages of the COVID-19 pandemic, pregnant women exhibited high rates of PTSD associated with the presence of state and trait anxiety, contact with a person diagnosed with COVID-19, and changes in their routine after the onset of the pandemic. As more than half of the included women reported PTSD symptoms, the findings of this study highlight the need for greater attention on the mental health of pregnant women during complex social situations such as the COVID-19 pandemic. The emotional effects of the pandemic on pregnant women who were receiving prenatal care during this period, which were generally caused by preventive measures such as social distancing, increased time spent at home, restrictions on in-person educational and recreational activities, and uncertainty about the duration of these restrictions and health and social issues, is significant. Considering that the negative effects of PTSD in pregnant women can also extend to the psychological development of offspring, as well as the emotional experience during subsequent pregnancies, understanding the process underlying the development of mental health disorders in pregnant women could be important during other public health emergencies or natural disasters, wherein greater attention should be paid to the specific vulnerability of pregnant women.

## Figures and Tables

**Table 1 ijerph-21-00461-t001:** Sociodemographic, clinical characteristics, and aspects related to the COVID-19 pandemic. (*n* = 149).

Characteristics	*n* (%)	Mean	Standard Deviation
Age (Years)		31.20	6.15
Marital status (With a partner)			
Without a partner	46 (30.8)		
With a partner	103 (69.2)		
Years of education			
≤8 years	17 (11.4)		
9 to 11 years	84 (56.4)		
≥12 years	48 (32.2)		
Living with children under 18			
None	47 (31.5)		
1–2 children	92. (61.7)		
≥3 children	10 (6.7)		
Living with elderly people > 60 years old			
None	126 (84.6)		
1 to 3	23 (15.4)		
Paid work (Yes)	65 (43.6)		
Faith or religious belief (Yes)	135 (90.6)		
Gestational trimester			
1st trimester	23 (15.4)		
2nd trimester	58 (38.9)		
3rd trimester	68 (45.6)		
Received health treatment (Yes)	73 (49.0)		
Currently using medication (Yes)	89 (59.7)		
Hospitalized in the last year (Yes)	37 (24.8)		
Psychiatric treatment prior to February 2020 (medication treatment) (Yes)	20 (13.4)		
Psychological support prior to February 2020 (Yes)	29 (19.5)		
Disease associated with current pregnancy (Yes)	102 (68.5)		
Flu symptoms in the last month (Yes)	41 (27.5)		
Contact with a suspected COVID-19 patient (Yes)	28 (18.8)		
Contact with a confirmed COVID-19 patient (Yes)	20 (13.4)		
Diagnosed with COVID-19 (Yes)	24 (16.1)		
Duration since COVID-19 diagnosis			
One week ago	14 (9.4)		
15 days ago	4 (2.7)		
A month or more ago	6 (4.0)		
Routine changed after the start of the COVID-19 pandemic (Yes)	132 (88.6)		
State anxiety			
None to moderate	62 (41.6)		
Severe	87 (58.4)		
Trait anxiety			
None to moderate	83 (55.7)		
Severe	66 (44.3)		

**Table 2 ijerph-21-00461-t002:** Distribution of demographic variables according to the presence of post-traumatic stress disorder (PTSD) ^#^ symptoms during the COVID-19 pandemic and associations between the variables and PTSD.

Variables	Impact Event Scale-Revised				*p*	OR (95% CI)
	<33		≥33			
	*n*	%	*n*	%		
Age (median)						
<32 years	31	40.8	45	59.2	0.296	Ref.
≥32 years	36	49.3	37	50.7		0.708 (0.370–1.353)
Marital status						
Without a partner	21	45.7	25	54.3	0.910 *	Ref.
With a partner	46	44.7	57	55.3		1.041 (0.518–2.092)
Years of education						
≤8 years	9	52.9	8	47.1	0.585 *	Ref.
9 to 11 years	39	46.4	45	53.6		1.298 (0.457–3.689)
≥12 years	19	39.6	29	60.4		1.717 (0.563–5.233)
Living with children under 18						
None	17	36.2	30	63.8	0.364 **	Ref.
1–2 children	45	48.9	47	51.1		0.592 (0.288–1.218)
≥3 children	5	50.0	5	50.0		0.567 (0.143–2.241)
Living with elderly people > 60 years old						
None	58	46.0	68	54.0	0.541 *	Ref.
1 to 3	9	39.1	14	60.9		1.327 (0.535–3.289)
Paid work						
No	39	46.4	45	53.6	0.683 *	Ref.
Yes	28	43.1	37	56.9		1.145 (0.597–2.198)
Faith or religious belief						
No	6	42.9	8	57.1	0.868 *	Ref.
Yes	61	45.2	74	54.8		0.910 (0.299–2.765)
Total	67	44.9	82	55.1		

^#^ The presence of PTSD was assessed using the Impact Event Scale-Revised. Having <33 points was considered indicative of the absence of PTSD, whereas having ≥33 points indicated the presence of PTSD. * Chi-square; ** Fisher’s Exact Test.

**Table 3 ijerph-21-00461-t003:** Distribution of gestational and health variables according to the presence of post-traumatic stress disorder (PTSD) ^#^ symptoms during the COVID-19 pandemic and associations between the variables and PTSD.

Variables	Impact Event Scale-Revised				*p*	OR (95% CI)
	<33		≥33			
	*n*	%	*n*	%		
Gestational trimester						
1st trimester	8	34.8	15	65.2	0.336 *	Ref.
2nd trimester	30	51.7	28	48.3		0.498 (0.183–1.354)
3rd trimester	29	42.6	39	57.4		0.717 (0.268–1.918)
Received health treatment						
No	38	50.0	38	50.0	0.208 *	Ref.
Yes	29	39.7	44	60.3		1.517 (0.792–2.905)
Currently using medication						
No	26	43.3	34	56.7		Ref.
Yes	41	46.1	48	53.9	0.742 *	0.895 (0.463–1.731)
Hospitalized in the last year						
No	52	46.4	60	53.6	0.532 *	Ref.
Yes	15	40.5	22	59.5		1.271 (0.598–2.702)
Psychiatric treatment prior to February 2020 (medication treatment)						
No	62	48.1	67	51.9		Ref.
Yes	5	25.0	15	75.0	0.054 *	2.776 (0.953–8.089)
Psychological support prior to February 2020						
No	57	47.5	63	52.5	0.206 *	Ref.
Yes	10	34.5	19	65.5		1.719 (0.738–4.003)
Disease associated with current pregnancy						
No	28	59.6	19	40.4	0.015 *	Ref.
Yes	39	38.2	63	61.8		2.381 (1.175–4.824)

^#^ The presence of PTSD was assessed using the Impact Event Scale-Revised. Having <33 points was considered indicative of the absence of PTSD, whereas having ≥33 points indicated the presence of PTSD. * Chi-square.

**Table 4 ijerph-21-00461-t004:** Distribution of anxiety variables and COVID-19 exposure and risk factors according to the presence of the presence of post-traumatic stress disorder (PTSD) ^#^ symptoms during the COVID-19 pandemic and associations between the variables and PTSD.

Variables	Impact Event Scale-Revised				*p*	OR (95% CI)
	<33		≥33			
	*n*	%	*n*	%		
Flu symptoms in the last month						
No	52	48.1	56	51.9	0.205 *	Ref.
Yes	15	36.6	26	63.4		1.610 (0.769–3.371)
Contact with a suspected COVID-19 patient						
No	59	48.8	62	51.2	0.053 *	Ref.
Yes	8	28.6	20	71.4		2.379 (0.973–5.817)
Contact with a confirmed COVID-19 patient						
No	65	50.4	64	49.6	0.001 *	Ref.
Yes	2	10.0	18	90.0		9.141 (2.037–41.010)
Diagnosed with COVID-19						
No	58	46.4	67	53.6	0.422 *	Ref.
Yes	9	37.5	15	62.5		1.443 (0.588–3.542)
Duration since COVID-19 diagnosis						
One week ago	5	35.7	9	64.3	0.848 *	Ref.
15 days ago	2	50.0	2	50.0		0.556 (0.059–5.241)
A month or more ago	2	33.3	4	66.7		1.111 (0.148–8.367)
Routine changed after the start of the COVID-19 pandemic						
No	14	82.4	3	17.6	0.001 *	Ref.
Yes	53	40.2	79	59.8		6.956 (1.906–25.386)
State anxiety						
None to moderate	44	71.0	18	29.0	<0.001 *	Ref.
Severe	23	26.4	64	73.6		6.802 (3.289–14.065)
Trait anxiety						
None to moderate	59	71.1	24	28.9	<0.001 *	Ref.
Severe	8	12.1	58	87.9		17.823 (7.404–42.901)

^#^ The presence of PTSD was assessed using the Impact Event Scale-Revised. Having <33 points was considered indicative of the absence of PTSD, whereas having ≥33 points indicated the presence of PTSD. * Chi-square.

**Table 5 ijerph-21-00461-t005:** Distribution of data according to factors associated with PTSD identified in the multiple binary logistic regression analysis.

Variables	ORaj	95% CI		*p*
		Lower Limit	Upper Limit	
State anxiety	2.71	1.11	6.57	0.028
Trait anxiety	12.20	4.60	32.34	<0.001
Routine changed after the start of the coronavirus pandemic	4.94	1.08	22.60	0.040
Contact with a confirmed COVID-19 patient	6.93	1.07	44.69	0.042

## Data Availability

Data available on request due toethical reasons. The data presented in this study are available on request from the corresponding author due to ethical reason.

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
