# Peer review of "Post-Traumatic Stress Disorder in Brazilian Pregnant Women at the Beginning of the Coronavirus Disease Pandemic"

_ijerph, 2024, doi:10.3390/ijerph21040461_

Round 1

Reviewer 1 Report

Comments and Suggestions for Authors

Well done

Author Response

We appreciate your considerations and suggestions. Considering, as mentioned, the importance of the topic and the need for a broader understanding regarding the associations between sociodemographic characteristics, mental health symptoms, and issues related to the COVID-19 pandemic in increasing the risk for PTSD symptoms, a research project is currently underway with the same sample, aiming for longitudinal follow-up.

Reviewer 2 Report

Comments and Suggestions for Authors

The manuscript has an intersting topic and somehow surprising results. I would suggest to revise the manuscript carefully considering the following points: 

Abstract: You write, that the prevalence of PTSD is 51% which is very high! Later you clarify that it means that 51% of the women have symptoms of PTSD. Please be correct here! Further results/ interpretation in the abstract has to be adapted after revision. 

Intro: l. 51-53: Why did you mention this? The Introduction would benefit from more information about the situation 2020 in Brazil, because it was diffent in the different countries. Was there a lock-down (when, how long?)? What about infection rates? 

l.87-90: What is your main research question? Do you have hypotheses? If not, please state clearly, what you are going to explore. 

Methods: l.99-100: you already wrote that some lines above.

I wonder if the snowball recruitment was a good idea: It might lead to selection bias. Women with the same background, similar experiences, jobs, neighbourhood etc. might have been included. This takes me to my next remark: 

A classical "Table 1" is missing providing participants' characteristics. There are only information about mean age, the others are sparse. A table would be helpful here. 

l. 119: "changes in routines" is too general for me, since the pandemic changed nearly everyone's lives. What does it mean/ what was the exact question? 

l.129-30: where is the cut-off from? It seems to be a bit arbitrary here. I later looked up the questionnaire and found the cut-off but it should be explained here. 

Statistics: Did you select confounding factors only by applying backward selections? Where there no theoretical considerations made?  

Results: 

In general, there is a big problem with causality: An association does not proof causality! Be careful here! In detail: 

l.173-75: Do pregnant women who receive psychological treatment really develop PTSD?? It doesn't make sense to me! But women with PTSD might more often visit a psychologist. The association is  clear and not surprising, however, the direction is unclear and is not the result of the regression! 

l.177-80: no interpretation in the results section; again: causality is not proven with these results. Further: a p-value of >.05 should be interpreted with caution. I would leave it out here- at least talk about a tendency or trend. 

l. 202 ff: pleas erevise the whole paragraph taking causality into account! 

Anxiety (state or  trait) might be a reason OR consequence of PTSD! 

It is wrong to say, that changes in daily routines cause PTSD- it can also be that PTSD changes my daily routines... 

Discussion: Please revise your discussion having causality in mind!!  Be exact with numbers: (l.224: 54.7% PTSD symptoms??)

Don't mix-up "prevalence of.. " with "symptoms of...". Have a look at all the literature you cite considering this! Because this might be a reason for this far-reaching difference of PTSD-prevalences. 

l.244-48: Maybe you should start the paragraph with this study. Other studies which do not confirm your results should follow. 

l.249 This is a limitation and should be mentioned in the strengths and limitations paragraph

l.268: Did you collect these data/ events or reasons for PTSD? This would have been interesting. If not it is a limitation. 

l.275: This paragraph is a bit confusing. Please fin a red line: what are potential reasons for PTSD in pregnancy? Which one have you identified (caution: causality!)? 

l. 337-245: This is not clearly shown in the results part/ has to be shown in the results, not in the discussion

l.355: "prevalence" or symptoms??  

Minor:

l.59: no comma after epidemics

Tab1. "years of study" --> better say years of education

"How many children do-..." there are two ** 

Tab.3: Flu symptoms the last month? or months? 

State/ Trait anxiety: delete"until" maybe write "none to moderate"

Tab4: Is ORaj a typo? 

Comments on the Quality of English Language

The manuscript needs to be read and revised by an English native speaker.

Author Response

Point-by-point response to Comments and Suggestions for Authors

Reviewer 2

Abstract:

You write that the prevalence of PTSD is 51%, which is very high! Later you clarify that it means that 51% of the women have symptoms of PTSD. Please be correct here! Further results/ interpretation in the abstract has to be adapted after revision.

Thank you very much for the observation. We agree with the comment and, to make it clearer to the reader, we changed the prevalence of PTSD to the presence of symptoms compatible with PTSD.

Intro:

  1. 51-53: Why did you mention this? The Introduction would benefit from more information about the situation 2020 in Brazil, because it was different in the different countries. Was there a lock-down (when, how long?)? What about infection rates?

We appreciate the comment and agree that presenting the panorama of the COVID-19 pandemic in 2020 in Brazil can offer the reader contextual elements to understand the importance of the research. Therefore, the quoted excerpt was replaced by:

“Since the first case of COVID-19 reported on February 25, 2020, Brazil has rapidly advanced to the highest number of cases in Latin America and the highest transmission rate in the world. Plagued by worsening economic, social, political and public health problems, at the end of 2020 the country accumulated 7,675,973 new cases and 194,949 deaths. The non-pharmacological measures adopted to contain the pandemic in Brazil were heterogeneous, with conflicting official guidelines on the use of masks. In the State of São Paulo, there was an official lockdown between March and August 2020, with a ban on public events and interruption of the main commercial and educational activities, remaining only the essential activities. However, adherence was not uniform throughout the region and became more flexible over time, especially for the most economically vulnerable population, who needed to work, as official emergency aid (approximately US$120 per family) was not enough nor extended to the entire population. There was also a crisis in the health system, with a shortage of hospital beds, especially for intensive care, and little availability of mechanical ventilators and oxygen. Furthermore, there was an underlying political crisis, following a heated electoral process a year earlier, which divided the popular opinion into two poles, with widespread political dissatisfaction.”

l.87-90:

What is your main research question? Do you have hypotheses? If not, please state clearly, what you are going to explore.

Thanks for the note. Making the research question clear is essential for reading the text from the authors' perspective. The text has been modified as follows:

“Therefore, based on the above so far, we hypothesize that the COVID-19 pandemic, due to its unprecedented and acute nature, has imposed significant challenges on risk groups, such as pregnant women, in particular due to the peculiarities of the emotional experiences lived during pregnancy, with potential risk mental health in general, including impacting the perception of events as potentially traumatic, increasing the risk of developing symptoms compatible with PTSD.”

Methods:

l.99-100: you already wrote that some lines above.

I wonder if the snowball recruitment was a good idea: It might lead to selection bias. Women with the same background, similar experiences, jobs, neighbourhood etc. might have been included. This takes me to my next remark:

A classical "Table 1" is missing providing participants'characteristics. There are only information about mean age, the others are sparse. A table would be helpful here.

We appreciate the comments. The recruitment method used could be a possible limitation, with perhaps a selection bias. However, it seems to us that it was the most appropriate approach for that moment in the pandemic, when the direct approach meant additional risk for this group. To reduce the risk of sampling bias, an invitation to participate in the study was also used through folders in the prenatal waiting room, which presented the research link through a QR Code. Thus, several patients were able to participate without being invited by another participant.

Regarding sample characterization data, Table 1 was included as suggested.

  1. 119:

"changes in routines" is too general for me, since the pandemic changed nearly everyone's lives. What does it mean/what was the exact question?

Thanks for the observation. When considering that the gestational experience already implies, to some extent, adapting to a series of transformations in which the pregnant woman does not exercise direct control, the intention was to better understand whether they identified the changes resulting from the pandemic specifically. The question asked was: Was your routine changed after the start of the pandemic?

l.129-30:

where is the cut-off from? It seems to be a bit arbitrary here. I later looked up the questionnaire and found the cut-off but it should be explained here.

Thank you for the consideration. The text was changed to: “The total score is the sum of the subscale scores. For this study, a score ≥33 points was considered indicative of risk for PTSD as established in a national15 and international16 study.

Statistics:

Did you select confounding factors only by applying backward selections? Where there no theoretical considerations made?

Thank you for your consideration. For the inclusion of variables in the multiple modeling, variables with p-values <0.20 were considered, along with those related to the outcome observed previously in the literature (theoretical framework). When conducting the modeling, in the event that a variable loses its significance or changes the direction of beta, or even enhances its effects, there would be an association among them to identify potential confounding factors or interactions. Even though it was a backward modeling technique, researchers generated more than one model, and ultimately, these were evaluated by the group to ensure that, at least, the assumptions of biological plausibility and strength of association were not violated.

Results:

In general, there is a big problem with causality: An association does not proof causality! Be careful here! In detail:

Thanks for the notes. We agree with the consideration and made changes to the writing of the text to make it even clearer that the analyzes present the associations between the variables without proposing a causal relationship between them.

l.173-75:

Do pregnant women who receive psychological treatment really develop PTSD?? It doesn't make sense to me! But women with PTSD might more often visit a psychologist. The association is clear and not surprising, however, the direction is unclear and is not the result of the regression!

We appreciate the relevant and necessary considerations to improve the results. The statements present in the quoted excerpt lead to a mistaken perception of a non-existent statistical fact and, for this reason, we modified the text.

“Table 2 presents the univariate analysis of the association between gestational and health variables with the presence of symptoms compatible with PTSD during the pandemic. Variables such as gestational age and others relating to the participants' general health status did not show a statistically significant difference between the groups with and without symptoms compatible with PTSD. However, the group of pregnant women who reported illness associated with the current pregnancy showed a statistically significant difference between the groups, where 61.8% belonged to the group with symptoms compatible with PTSD.”

l.177-80:

no interpretation in the results section; again: causality is not proven with these results. Further: a p-value of >.05 should be interpreted with caution. I would leave it out here- at least talk about a tendency or trend.

We thank you for your consideration and removed the excerpt as suggested.

  1. 202 ff:

please revise the whole paragraph taking causality into account!

Anxiety (state or trait) might be a reason OR consequence of PTSD!

It is wrong to say, that changes in daily routines cause PTSD- it can also be that PTSD changes my daily routines...

We thank you for your comments and made some changes to clarify the association between the variables.

“A multiple binary logistic regression model created after univariate logistic regression analysis revealed that state and trait anxiety, changes in routine due to the pandemic and contact with a patient confirmed with COVID-19 were factors associated with the presence of symptoms compatible with PTSD (Hosmer-Lemeshow = 0.868). One of the aspects that deserves to be highlighted is the result of the trait anxiety variable, where there was a 12-fold increase in presenting symptoms compatible with PTSD (adjusted OR = 12.20; 95% CI, 4.60 – 32.34).”

Discussion:

Please revise your discussion having causality in mind!! Be exact with numbers: (l.224: 54.7% PTSD symptoms??)

Don't mix-up "prevalence of.. " with "symptoms of...". Have a look at all the literature you cite considering this! Because this might be a reason for this far-reaching difference of PTSD-prevalences.

Thanks for the guidance. Changes were made to the text as requested.

l.244-48: Maybe you should start the paragraph with this study. Other studies which do not confirm your results should follow.

Thanks for the suggestion. The changes were made as follows:

“In a Turkish study pregnant women conducted during the COVID-19 pandemic using the IES-R, present with prevalence data stratified according to symptom severity, 55.4% of the women exhibited PTSD symptoms (cutoff ≥33 points), which, according to the authors, indicated a moderate to severe symptom impact of the pandemic, a finding comparable with the results of the present study31. In another study, of Jordanian pregnant women conducted using the IES-R which was also used in the present study, 58.6% of the pregnant women reported PTSD symptoms, a rate higher than that observed by us30. However, the authors used a lower cutoff score (22 points) than that used the present study (33 points); thus, the results of both studies may not be comparable.”

l.249 This is a limitation and should be mentioned in the strengths and limitations paragraph

Thanks for the note. It will be mentioned in the paragraph regarding limitations.

l.268: Did you collect these data/ events or reasons for PTSD? This would have been interesting. If not it is a limitation.

Thank you for the recommendation. The data that could be collected at the time of the evaluation were the presence of clinical complications. The other variables were not collected, mainly due to the cross-sectional study, making it impossible to monitor throughout the pregnancy. This will be mentioned in the paragraph about the limitations of the study.

l.275: This paragraph is a bit confusing. Please fin a red line: what are potential reasons for PTSD in pregnancy? Which one have you identified (caution: causality!)?

We appreciate the contribution. To reorganize the reasoning, we changed its presentation in the text, seeking a better division and better reorganization of the exposed themes.

  1. 337-245: This is not clearly shown in the results part/ has to be shown in the results, not in the discussion

We appreciate the guidance. The data was included with greater clarity in the results section and the text was reorganized in the discussion section.

l.355: "prevalence" or symptoms??

Thanks for the comment. Correction was carried out.

Minor:

l.59: no comma after epidemics

Adjusted.

Tab1. "years of study" --> better say years of education

Adjusted.

"How many children do-..." there are two **

Adjusted.

Tab.3: Flu symptoms the last month? or months?

The question was about flu symptoms in the last month.

State/ Trait anxiety: delete "until" maybe write "none to moderate"

Adjusted.

Review Report Form

Quality of English Language

Moderate editing of English language required.

The manuscript needs to be read and revised by an English native speaker.

We were surprised by the observation in relation to English, as the text had already been subjected to a translation from Portuguese to English through a reliable firm, Taylor & Francis. However, following your observation, we will send the final text for further English review, requesting a native professional from the USA.

Reviewer 3 Report

Comments and Suggestions for Authors

The literature is extensive and most of the items come from 2020, I suggest adding the latest publications on the presented topic, from 2021-2024. The authors of the publication indicated that the respondents included pregnant women with clinical complications. Whether? And if so, how could clinical complications have influenced the subjects' PTSD levels (apart from stress caused by Covid-19)? What exactly were the criteria for selecting respondents?

Author Response

Thanks for the recommendation. We added five new articles, published between 2021 and 2024, thus updating the bibliography, as requested.

Yes, the sample was composed of low and high obstetric risk pregnant women. The univariate analysis, considering variables such as previous medical treatment or use of medications, among other health-related variables, did not demonstrate a statistically significant difference between the groups.

To be included in the present study, the pregnant woman needed to consent to participate in the research, and undergo prenatal care at one of the two university hospitals that hosted the study, in São Paulo, Brazil.
